# Effect of the *Lactococcus Lactis* 11/19-B1 Strain on Atopic Dermatitis in a Clinical Test and Mouse Model

**DOI:** 10.3390/nu12030763

**Published:** 2020-03-14

**Authors:** Takato Suzuki, Kyoko Nishiyama, Koji Kawata, Kotaro Sugimoto, Masato Isome, Shigeo Suzuki, Ruriko Nozawa, Yoko Ichikawa, Yoshihisa Watanabe, Tatsuo Suzutani

**Affiliations:** 1Department of Microbiology, School of Medicine, Fukushima Medical University, Fukushima 960-1295, Japan; tm1451@fmu.ac.jp (T.S.); kyoko@fmu.ac.jp (K.N.); 2Laboratory Animal Research Center, School of Medicine, Fukushima Medical University, Fukushima 960-1295, Japan; kkawata@fmu.ac.jp; 3Department of Basic Pathology, School of Medicine, Fukushima Medical University, Fukushima 960-1295, Japan; sugikota@fmu.ac.jp; 4Isome Children’s Clinic, Fukushima 960-8165, Japan; misome@cd6.so-net.ne.jp; 5Department of Pediatrics, Ohara General Hospital, Fukushima 960-8611, Japan; s_shigeo@ohara-hp.or.jp; 6Department of Pediatrics, Fujita General Hospital, Kunimi, Date, Fukushima 969-1793, Japan; rurinozawa@qmail.com; 7Ichikawa Clinic, Fukushima 960-0112, Japan; i-clinic@d4.dion.ne.jp; 8Tohoku Kyodo Milk Co., Ltd., Motomiya, Fukushima 969-1104, Japan; watanabe_zen@tk-holstein.com

**Keywords:** atopic dermatitis, Lactococcus lactis, clinical test, mouse model, t cell subsets

## Abstract

Some lactic acid bacteria (LAB) are known to improve atopic dermatitis (AD) through the regulation and stimulation of the host immune system. In this study, we found that ingestion of yogurt containing *Lactococcus lactis* 11/19-B1 strain (*L. lactis* 11/19-B1) daily for 8 weeks significantly improved the severity scoring of atopic dermatitis (SCORAD) system score from 38.8 ± 14.4 to 24.2 ± 12.0 in children suffering from AD. We tried to identify which LAB species among the five species contained in the test yogurt contributed to the improvement in AD pathology using an AD mouse model induced by repeated application of 1-fluoro-2, 4-dinitrobenzene (DNFB). AD-like skin lesions on the dorsal skin and ear were most improved by *L. lactis* 11/19-B1 intake among the five LAB species. In addition, analysis of CD4+ T cell subsets in Peyer’s patches (PPs) and cervical lymph nodes (CLNs) indicated that the intake of *L. lactis* 11/19-B1 generally suppressed all subsets related to inflammation, i.e., Th1, Th2 and Th17, instead of activating the suppressive system, Treg, in the AD mouse model. Histological observations showed ingestion of *L. lactis* 11/19-B1 significantly suppressed severe inflammatory findings, such as inflammatory cell filtration, epidermal erosion and eosinophil infiltration. These results suggest that the immunomodulatory effects of *L. lactis* 11/19-B1 contribute to improvements in AD pathology.

## 1. Introduction

Atopic dermatitis (AD) is a chronic inflammatory skin disease characterized by repeated episodes of exacerbation and remission. Many AD patients have an atopic predisposition, and frequent skin damage associated with itching results in a decreased quality of life [1]. The etiology of AD involves the disruption of skin barrier function and immune abnormalities, which influence each other and affect the pathogenesis and progression of the disease. In terms of the immunological abnormalities of AD, in particular T cell subset composition, such as Th1, Th2 and Th17, and changes in subset-specific cytokines participate in the genesis of the disease. Previous studies have shown that Th2 cells elicit preferential immune responses in the acute phase and Th1 cells in the chronic phase of AD [2,3]. In addition, Th17 cells, which play an important role in the pathogenesis of psoriasis and production of certain cytokines from keratinocytes, were also observed in the peripheral blood of AD patients [4]. Therefore, it is speculated that the control of T cell function and composition in AD might contribute to alleviation of the pathological condition.

Recently, the immunoregulatory functions of lactic acid bacteria (LAB), such as enhancement of natural killer (NK) cell activity and antibody production and control of cytokine production, have attracted a good deal of attention [5,6]. It has been reported that some LAB strains, such as the *Lactobacillus rhamnosus* GG strain [7,8] and the *Lactobacillus paracasei* KW3110 strain [9,10], contribute to improvements in AD pathology through regulation of the Th1/Th2 balance and anti-inflammatory response. However, different functions were observed in LAB strains, even in the same species.

The *Lactococcus lactis* 11/19-B1 strain (*L. lactis* 11/19-B1) isolated from the surface of kiwi fruit activates innate immunity and enhances tolerance against *Pseudomonas aeruginosa* infection in silkworms compared to other LAB strains [11,12]. Moreover, we have demonstrated that ingestion of yogurt containing this strain decreases low density lipoprotein (LDL) levels and activates cellular immunity in humans [13]. These observations suggest that *L. lactis* 11/19-B1 might improve AD pathology through regulation and stimulation of the host immune system. Therefore, we studied whether the ingestion of *L. lactis* 11/19-B1 could improve the clinical symptoms of AD patients by means of a clinical test in this study. Furthermore, we investigated the anti-allergic activities and related mechanisms of *L. lactis* 11/19-B1 using a mouse AD model induced by the repeated application of 1-fluoro-2, 4-dinitrobenzene (DNFB).

## 2. Materials and Methods 

### 2.1. Subjects and Study Design

The study was conducted according to the guidelines set out in the Declaration of Helsinki and all procedures involving human subjects were approved by the Ethics Committee of Fukushima Medical University (Approval no. 2061). The subjects, aged two to 15 years, with AD, were recruited from the Isome Children’s Clinic, Ichikawa Clinic, Ohara General Hospital Department of Pediatrics and Fujita General Hospital Department of Pediatrics. Written informed consent was obtained from each subject’s parents prior to enrollment in this study. 

The subjects, of whom characteristics are shown in Table 1, ingested 80g of the yogurt daily for 8 weeks after a four-week pre-observation period (Figure 1A). During the study period, subjects were not restricted in terms of AD treatment, but were asked not to eat foods containing other lactic acid bacteria. All volunteers received topical steroids and one or more additional drugs (Table 1). Almost all subjects continued on these treatments during the study, but a few reduced drug use late in the yogurt-intake period due to improvements in symptoms.

The AD severity assessment and blood sampling were performed every 4 weeks from the start to end of the test including the pre-observation and yogurt ingestion periods. The severity of AD was evaluated by the severity scoring of atopic dermatitis (SCORAD) system. The SCORAD soring ranges from zero to 103 and is classified as mild (0–25 point), moderate (26–50 point), or severe (>50 point) [14]. Each blood sample was analyzed for blood cell counts, aspartate transaminase (AST), alanine transaminase (ALT), alkaline phosphatase (ALP), total bilirubin, creatinine, immunoglobulins (IgG, IgM and IgE), thymus and activation-regulated chemokine (TARC) and C-reactive protein (CRP) in the laboratory of each clinic or a commercial laboratory.

### 2.2. Yogurt

The yogurt fermented with the Streptococcus thermophilus ST-20 strain (S. thermophilus), Lactobacillus delbrueckii subspecies bulgaricus LB-12 strain (L. bulgaricus), Lactobacillus acidophilus La-5 strain (L. acidophilus), Bifidobacterium lactis Bb-12 strain (B. lactis) and the L. lactis 11/19-B1 strain, was produced once a week by Tohoku Kyodo Milk Industry Co., Ltd. (Fukushima, Japan) and sent to subjects at 4 °C.

### 2.3. Animals and Diets

All experiments using mice were performed in accordance with the institutional guidelines of Fukushima Medical University for the care and use of animals (Approval no. 2019057).

Eight-week-old female Balb/c mice were purchased from Charles River Japan and maintained in a specific pathogen-free animal facility at Fukushima Medical University. Mice had ad libitum access to water and AIN-93G diet (Clea, Tokyo, Japan) with or without one of the heat-killed LAB strains at 0.05%. 

The diets containing the LAB strains were prepared as follows. The six LAB strains tested in this study consisted of the five strains from the tested yogurt and the *Lactococcus lactis* JCM20101 strain (*L. lactis* JCM20101), purchased from the Japan Collection of Microorganisms (RIKEN BRC, Tsukuba, Japan). These LAB strains were cultured at 37 °C for 24–48 h in MRS broth (BD Difco, MD, USA) or Brucella broth (BD Difco, MD, USA). Cultured bacteria were washed twice with 0.85% NaCl, suspended in sterilized distilled water, heat-killed at 100 °C for 30 min, and lyophilized. Each dried LAB strain was then mixed with AIN-93G and irradiated with γ-rays.

### 2.4. AD Mouse Model

A mouse model of AD was produced according to the method of Hussain et al. with modifications as described in Figure 2A [15]. Briefly, on day 14 after the start of experimental feeding of eight-week-old female BALBc/A mice, the dorsal region of the mice was shaved. One hundred µL of 0.15% DNFB (Sigma-Aldrich, Tokyo, Japan) in acetone/olive oil (3:1) was applied to the shaved dorsal skin of the mice on days 15 and 19. On days 23, 25, and 27, 0.2% DNFB was applied to the dorsal skin (100 µL) and both ears (25 µL each). Tissue samples and blood were collected from mice anesthetized with 2% isoflurane (Intervet, Tokyo, Japan) at 6 h after the last 0.2% DNFB application (day 27). Control mice (vehicle) were sensitized and reapplied with acetone/olive oil (3:1) only. 

The severity of AD-like skin lesions was evaluated at 24 h after repeated DNFB application on day 25 based on four criteria: (1) erythema/hemorrhage, (2) dryness/scaling, (3) edema/swelling and (4) erosion/excoriation. Symptom severity was scored as 0 = none, 1 = mild, 2 = moderate, or 3 = severe, and the sum of the individual scores was then taken as the dermatitis index (ADI) [15].

AD-like skin reactions were assessed by measuring the ear swelling induced by repeated DNFB applications (day 23, 25, and 27) using a micrometer (Mitutoyo, Kanagawa, Japan). The measurement was performed on the left and right ears before (0 h) and after DNFB application (1 h, 6 h). Ear swelling was calculated by the difference between the indicated time points and 0 h.

### 2.5. Measurement of Total IgE in Plasma

The plasma concentration of total IgE was measured using a Mouse IgE ELISA kit (Bethyl Laboratories, Texas, and USA) following the manufacturer’s protocol.

### 2.6. Flow Cytometric Analysis (FACS)

Cells derived from Peyer’s patches (PPs) and cervical lymph nodes (CLNs) were stained with specific surface markers (CD3, CD4) for flow cytometry using fluorescence-conjugated anti-mouse monoclonal antibodies (Abs); i. e., PE-Cy7-anti-CD3 (145-2C11) and Pacific blue-anti-CD4 (RM4-5) Abs (BD Biosciences, NJ, USA). To analyze the cells expressing cytokines IFN-γ, IL-4 and IL-17, intracellular staining was performed using Intracellular Fixation & Permeabilization Buffer (eBioscience, CA, USA) and PerCP-Cy5.5-anti-IFN-γ (XMG1.2), PE-anti-IL-4 (11B11) and APC-anti-IL-17A (17B7) Abs (eBioscience, CA, USA) after stimulation for 16 h with a Cell Stimulation Cocktail (Invitrogen, MD, USA) as described in by the manufacturer. Furthermore, transcription factor FoxP3 was stained by intracellular staining using APC-anti-FoxP3 (FJK-16s) Ab according to the protocol included with the Foxp3/Transcription Factor Staining Buffer Set (eBioscience, CA, USA). The cells were analyzed using a FACS Canto II system (BD Biosciences). In each experiment, specimens were analyzed for singlet events with doublet discrimination. All data were examined by FACSDiva software (BD Biosciences, NJ, and USA) and Flowjo software (Tree Star, OR, USA).

### 2.7. Histological Evaluation

Dorsal skin and ear samples were fixed with 10% buffered neutral Formalin solution, paraffin-embedded, sectioned (3 µm), and subsequently stained with hematoxylin and eosin (HE) or an Eosinophil–Mast cell stain kit (Scy Tek Laboratories, UT, USA). HE-stained sections were evaluated using a semi-quantitative scoring system based on two criteria: (1) inflammatory cell infiltration and (2) epidermal erosion, using a light microscope (Olympus, Tokyo, Japan). The severity was scored as 0 = none, 1 = mild, 2 = moderate, or 3 = severe. Three sections from individual mice were blindly examined by a pathologist and the sum of the individual scores was compared between each group.

The Eosinophil–Mast cell stain kit was used in accordance with the manufacturer’s protocol for eosinophil visualization. The number of eosinophils was counted in five high-power fields (HPF) per slide, and their average was compared between each group [16].

### 2.8. Statistics

All data were expressed as means ± SD. For statistical testing of normality, the Kolmogorov–Smirnov test was used. Statistical significance was assessed by a Mann–Whitney U test for single comparisons and by a Steel–Dwass test after a Friedman test for multiple comparisons, with *p* ≤ 0.05 considered significant.

## 3. Results

### 3.1. Effect of L. Lactis 11/19-B1-Containing Yogurt Intake in AD Patients

The characteristics of the subjects at the start of the study are summarized in Table 1.

To investigate the effects of the yogurt in improving AD, AD patients ingested 80g of the test yogurt daily for 8 weeks after a 4-week pre-observation period. SCORAD scores in AD patients were decreased, but not significantly, at 4 weeks after the start of yogurt intake compared to those at the pre-observation period. At 8 weeks, significant improvements were observed (Figure 1B,C). On the other hand, biomarkers IgE, TARC, LDH and EOS, which reflect the pathology of AD, and other biochemical data did not differ between before and after the test yogurt intake. Thus, the contribution of test yogurt ingestion to the improvement in AD pathology was confirmed, but the underlying mechanisms remained unknown.

### 3.2. Identification of LAB Strains Inducing Improvement in AD Pathology Using a Mouse Model

Since intake of the test yogurt improved the clinical symptoms of AD in patients, we tried to identify the LAB species from among the five species contained in the test yogurt that contributed to the improvement in AD pathology using a mouse AD model induced by the repeated application of DNFB (Figure 2A). The AD model mice developed severe dermatitis on DNFB application, but no symptoms were observed in the vehicle group (Figure 2B). The suppressive effect on dermatitis in LAB intake mice groups was observed only in those fed the *L. lactis* 11/19-B1 and JCM20101 strains.

The time course of ear swelling revealed that the atopic group showed significant ear swelling at 1 and 6 h after the last DNFB application compared to the vehicle group (Figure 2C). The ear swelling was significantly suppressed in the *L. lactis, L. bulgaricus, B. lactis* and *S. thermophilus* intake groups at 6 h, but only the mice fed *L. lactis* 11/19-B1 or JCM20101 strains showed a positive effect at 1 h.

Plasma total IgE was increased significantly in the atopic group compared to that in the vehicle group. The intake of two *L. lactis* strains improved AD symptoms and ear swelling, but a suppressive effect on the plasma IgE value was observed only in the *L. acidophilus*-intake group (Figure 2D).

These results suggested that the *L. lactis* 11/19-B1 contributed the most to the suppression of AD symptoms in the human clinical test. Therefore, we examined the effect of *L. lactis* 11/19-B1 on AD pathology in greater detail through a comparison with *L. bulgaricus* (as a negative control) in subsequent experiments.

### 3.3. Altered Distribution of CD4+ T Cell Subsets with Ingestion of L. Lactis 11/19-B1

To investigate whether the ingestion of *L. lactis* 11/19-B1 had an effect on the distribution of CD4+ T cell subsets in DNFB-induced AD mice, we analyzed a proportion of T cell subsets in monocyte samples isolated from Peyer’s patches (PPs) and cervical lymph nodes (CLNs) by FACS.

The atopic group had significantly increased numbers of IFN-γ–producing cells (Th1 cells), IL-4–producing cells (Th2 cells) and Th17A-producing cells (Th17 cells) in the PPs and CLNs compared to the vehicle group (Figure 3). In the *L. lactis* 11/19-B1 group, Th1 and Th17 cells were significantly decreased in the PPs, and Th1, Th2 and Th17 cells in the CLNs compared to the atopic group. In contrast, the *L. bulgaricus* group significantly reduced the number of Th1 cells in CLNs compared to the atopic groups. No changes in Treg cells were observed in any of the groups. These results indicated that the intake of *L. lactis* 11/19-B1 generally suppressed all subsets related to inflammation, i.e., Th1, Th2 and Th17, instead of activating of the suppressive system, Treg, in the AD mouse model.

### 3.4. Histological Changes in AD Pathology Associated with the Ingestion of L. Lactis 11/19-B1

Histological evaluation of the dorsal skin and ears in the atopic group showed severe inflammatory findings including inflammatory cell infiltration, epidermal erosion and eosinophil infiltration compared to the vehicle group (Figure 4 and Figure 5). On the other hand, the *L. lactis* 11/19-B1 group showed significantly suppressed inflammatory findings for the dorsal skin and ears compared to the atopic group. These effects were not observed in the *L. bulgaricus* group. This result indicated that ingestion of *L. lactis* 11/19-B1 relieved inflammation induced by the repeated application of DNFB and contributed to improvements in AD pathology.

## 4. Discussion

Lactic acid bacteria (LAB) are familiar microorganisms used for the production of fermented foods such as yogurt and probiotic foods. It has been reported that the intake of LAB has beneficial effects on diseases such as *Helicobacter pylori* infection [17,18], viral infection [19,20], allergic disease [21], inflammatory bowel disease [22] and cancer [23]. In particular, immunoregulatory functions, such as NK cell activation, enhancement of antibody production and control of cytokine production associated with the ingestion of LAB, have attracted a good deal of attention, and various species and strains with immunologically different effects have been reported [5,6]. Since many immune cells and cytokine responses are involved in the pathology of AD, the ingestion of LAB may regulate the host immune system and contribute to the alleviation of symptoms. Therefore, we studied the effects of the ingestion of *L. lactis* 11/19-B1-containing yogurt (test yogurt) on AD patients and in a mouse model.

First, we observed that the ingestion of the test yogurt decreased SCORAD scores in AD patients in an ingestion period-dependent manner (Figure 1). During the study, the subjects continued their drug treatment, but none began additional treatment. Therefore, we believe that the remission of the symptoms was caused by the yogurt intake. Previous studies reported that some LAB strains, such as *Lactobacillus rhamnosus* GG strain [7,8] and *Lactobacillus paracasei* KW3110 strain [9,10], contribute to improvements in AD pathology through the regulation of Th1/Th2 balance and anti-inflammatory responses. However, there were no changes in blood biomarkers (IgE, TARC, LDH and Eosinophil) between patients pre- and post-intake of the yogurt in our clinical study; therefore, the detailed mechanism remained unclear. The test yogurt was fermented with the five LAB strains (*L. lactis* 11/19-B1, *L. bulgaricus, L. acidophilus, B. lactis* and *S. thermophilus*). Therefore, we tried to identify the species that contributed to the improvement in AD pathology by using a mouse AD model and determined that only *L. lactis* 11/19-B1 was related to the improvement. 

A single application of DNFB to sensitized mice is known to induce Th1-dominant delayed-type hypersensitivity [24], while repeated application of DNFB shifts the immune response from a Th1-dominant response to a Th2-dominant immediate-type hypersensitive response or a Th1/Th2 mixed response [25,26]. Previous studies have shown that the repeated application of DNFB to sensitized mice induces some AD-like pathologies, such as increased IgE, eosinophil infiltration, and increased Th2 cells at the lesion site [27,28,29]. In this study, only the ingestion of *L. lactis* 11/19-B1 significantly improved the AD-like skin responses induced by DNFB (Figure 2). 

Previous studies have also shown that interferon-α (IFN-α) produced by mainly plasmacytoid dendritic cells (pDC) induces Th1 cells and inhibits IgE production [30,31]. These effects may contribute to improvements in AD pathology. *L. lactis* JCM20101 and JCM5805 strains were reported to activate pDC and induce IFN-α production [32]. However, IFN-α production was observed only in the *L. lactis* JCM20101 used as a positive control and not in other *L. lactis* strains. Therefore, it was suggested that *L. lactis* 11/19-B1 contributes to improvements in AD pathology by mechanisms other than IFN-α production.

It is known that LAB taken into the body makes contact with the immune system of the gut-associated lymphoid tissues (GALTs). PPs, one of the GALTs, induce a general immune response through antigen uptake by M cells covering the PPs. It was reported that lymphocytes from the PPs enter systemic circulation after being released from the intestine [33,34]. Therefore, modification of the intestinal immune system associated with the ingestion of LAB may affect the regulation of the systemic immune system. In this study, ingestion of *L. lactis* 11/19-B1 produced a decrease in the percentage of Th1, Th2 and Th17 cells among the CD4+ T cells in the GALT (PPs) or lymph nodes at the AD lesion site (CLNs) (Figure 3). In addition, *L. lactis* 11/19-B1 suppressed dorsal skin and ear inflammatory findings associated with the repeated application of DNFB (Figure 4 and Figure 5). Previous studies have shown that the CD4+ T cell subset composition is involved in the genesis of AD pathology, and that subset regulation may contribute to an improved AD pathology [2,3,4]. In fact, the ingestion of *Lactobacillus casei* Shirota or *Enterococcus faecalis* EF-2001 elicits an ameliorating effect on AD pathology associated with improved T cell subset composition [35,36]. These reports are consistent with our results and support our findings that the ingestion of *L. lactis* 11/19-B1 induces amelioration of AD pathology through modulation of the immune response.

On the other hand, the increase in plasma total IgE observed with the repeated application of DNFB did not change with the ingestion of *L. lactis* 11/19-B1 (Figure 2D). In AD patients, total IgE is frequently elevated and is known to correlate with disease severity [37]. So far, it has been considered that IgE production is controlled by IL-4 derived from Th2 cells. However, Th2 cells are localized at the germinal center (GC) where antibody production occurs. Recent studies have reported the involvement of follicular T cells (Tfh cells), which were newly discovered in T cells, are localized in the GC and play an important role in B cell proliferation, selection and maturation [38,39,40]. Tfh cells produce IL-4 through a different mechanism than that of Th2 cells, and the production of IgE is dependent on Tfh cell-derived IL-4 [41,42]. Based on these reports, our results suggest that the ingestion of *L. lactis* 11/19-B1 is not involved in Tfh cell-regulated IgE production. In addition, it was observed in this study that eosinophil infiltration into lesions controlled by Th2 cells was significantly suppressed by *L. lactis* 11/19-B1 intake (Figure 5). These results support the notion that the alteration of IL-4 on CD4+ T cells observed in this study contributed to the number of Th2 cells and suppressed the AD pathology. 

Previous studies have shown that LAB, whether live or dead, have immunomodulatory effects in the host [9,10]. In this study, human AD patients were fed live LAB and mice were fed dead LAB, but improvements in AD were observed in both experiments. Therefore, the ingestion of *L. lactis* 11/19-B1 appears to be effective in improving the AD pathology, regardless of whether live or dead. We plan to continue our study of the mechanism underlying the immunomodulation function of *L. lactis* 11/19-B1.

## 5. Conclusions

Our study demonstrated that the ingestion of *L. lactis* 11/19-B1-containing yogurt reduces SCORAD scores in AD patients. Furthermore, the intake of dead *L. lactis* 11/19-B1 improves AD pathology in a mouse model through the suppression of the percentage of Th1, Th2 and Th17 in T cells and the inhibition of eosinocyte infiltration into the AD lesion site. These results indicated that *L. lactis* 11/19-B1 is a promising candidate as a functional food to alleviate atopic dermatitis.

## Figures and Tables

**Figure 1 nutrients-12-00763-f001:**
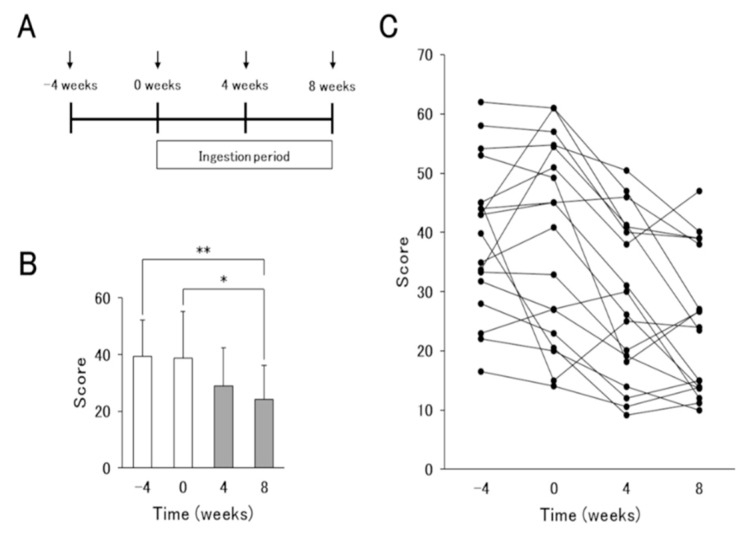
Results of the severity scoring of atopic dermatitis (SCORAD) index before and after *L. lactis* 11/19-B1-containing yogurt ingestion for 8 weeks. (**A**) Schematic representation of the experiment. Arrows indicate the times of the severity assessment and blood sampling in atopic dermatitis (AD) patients. (**B**) Average and (**C**) changes in individual volunteers in the SCORAD index associated with the ingestion of *L. lactis* 11/19-B1-containing yogurt are shown. Results are expressed as the means ± SD of 18 independent subjects. * *p* < 0.05, ** *p* < 0.01 by a Steel–Dwass test for multiple comparisons.

**Figure 2 nutrients-12-00763-f002:**
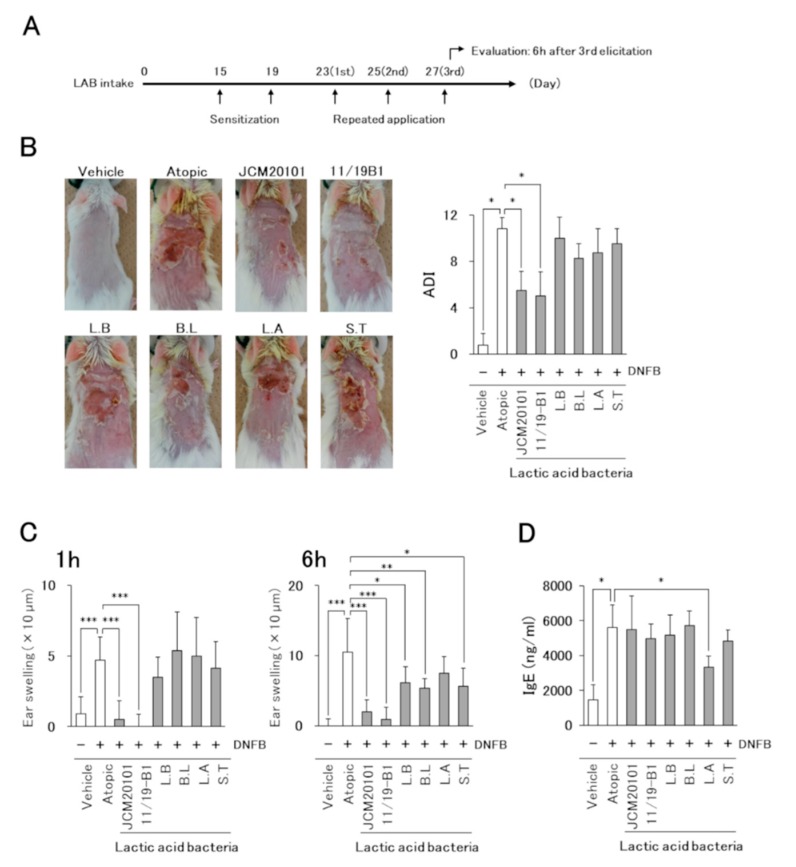
Effects of *L. lactis* 11/19-B1 intake on the development of 1-fluoro-2, 4-dinitrobenzene (DNFB)-induced AD-like symptoms. (**A**) Schematic representation of the experiment. (**B**) The severity of AD-like skin lesions was evaluated at 24 h after the repeated application of DNFB on day 25. Images are from a single experiment and are representative of each group (*n* = 4). (**C**) Ear swelling at 1 and 6 h after the last DNFB application (day 27) is shown. (**D**) The blood samples were collected at 6 h after the last DNFB application, and plasma total IgE was measured by ELISA. Results are expressed as the means ± standard deviation (SD) of four determinations for each group (*n* = 4). * *p* < 0.05, ** *p* < 0.01, *** *p* < 0.001 by Mann–Whitney U test. 1-fluoro-2, 4-dinitrobenzene (DNFB), vehicle group (Vehicle), atopic group (Atopic), lactic acid bacteria (LAB), *L. lactis* JCM20101 (JCM20101), *L. lactis* 11/19-B1 (11/19-B1), *L. bulgaricus* (L.B), *B. lactis* (B.L), *L. acidophilus* (L.A), *S. thermophilus* (S.T).

**Figure 3 nutrients-12-00763-f003:**
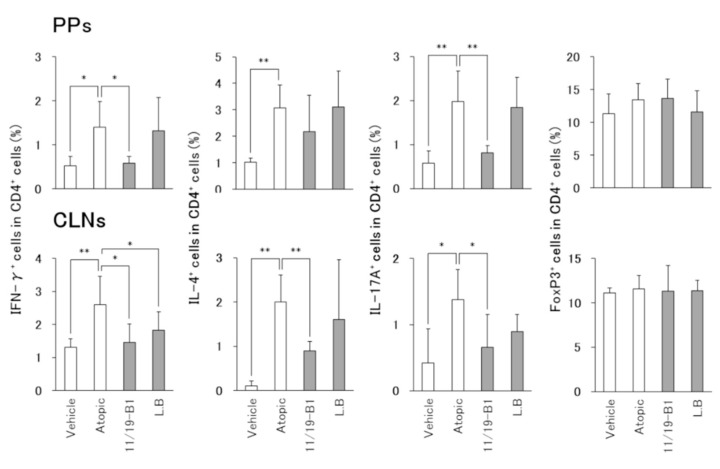
Altered composition of CD4+ T cell subsets in total CD4+ T cells associated with *L. lactis* 11/19-B1 intake in DNFB-induced mice. Changed composition of CD4+ T cell subsets in the PPs and CLNs collected at 6 h after the last DNFB application were measured by intracellular staining and FACS analysis. Results are expressed as the means ± SD of five determinations for each group (*n* = 5). * *p* < 0.05, ** *p* < 0.01 by Mann–Whitney U test. Peyer’s patches (PPs), cervical lymph nodes (CLNs). *L. bulgaricus* (L.B).

**Figure 4 nutrients-12-00763-f004:**
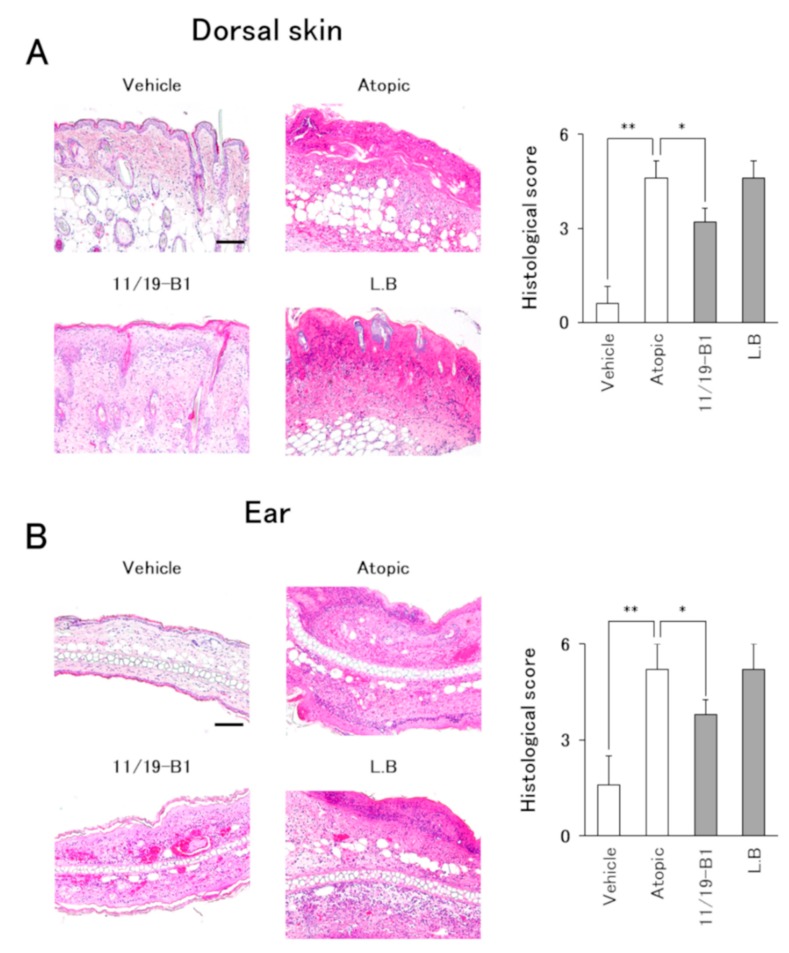
Evaluation of histological severity associated with *L. lactis* 11/19-B1 intake in DNFB-induced mice. Dorsal skin and ear samples were collected at 6 h after the last DNFB application, and subsequently stained with hematoxylin and eosin (HE). The sum of the individual histological severity scores for the dorsal skin (**A**) and ears (**B**) was compared for each group. Results are expressed as the means ± SD of five determinations for each group (*n* = 5). Images are from a single experiment and are representative of each group. Original magnification was 100× (scale bar, 100 µm). * *p* < 0.05, ** *p* < 0.01 by Mann–Whitney U test. *L. bulgaricus* (L.B).

**Figure 5 nutrients-12-00763-f005:**
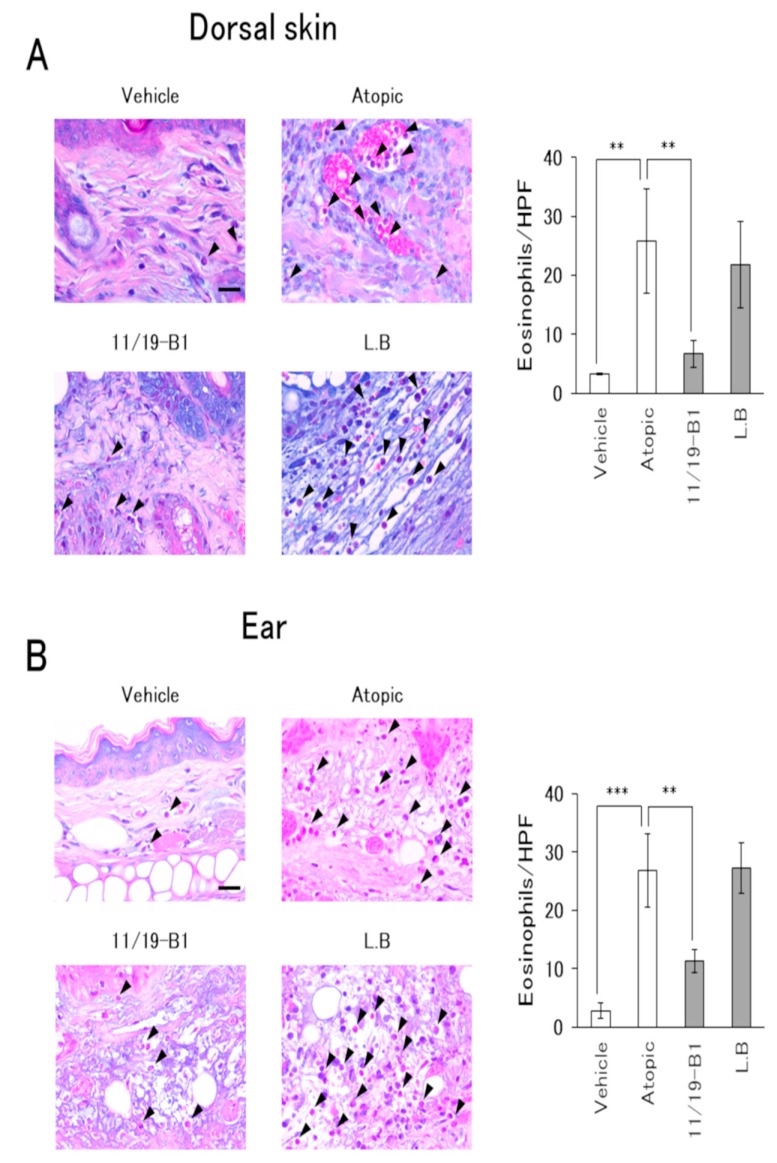
Evaluation of eosinophil infiltration associated with *L. lactis* 11/19-B1 intake in DNFB-induced mice. Dorsal skin (**A**) and ear (**B**) samples were collected at 6 h after the last DNFB application, and subsequently stained with an Eosinophil–Mast cell stain kit. The number of eosinophils in five high-power fields per slide was compared for each group. Arrows indicate eosinophils. Results are expressed as the means ± standard deviation (SD) of five determinations for each group (*n* = 5). Images are from a single experiment and are representative of each group. Original magnification was 400× (scale bar, 20 µm). ** *p* < 0.01, *** *p* < 0.001 by Mann–Whitney U test. *L. bulgaricus* (L.B).

**Table 1 nutrients-12-00763-t001:** Summary of subjects.

	Atopic Group
*N*	18
Age	8.4 ± 3.6
Sex (% of male)	66.7
SCORAD index	38.2 ± 16.4
IgE (mg/dL)	1757.4 ± 1724.3
TARC (pg/mL)	1246.0 ± 1206.0
Treatment (%)	
Tropical steroid	100
Heparinoid	83.3
Antihistamines	55.6
Tacrolims	16.7

Data are presented as the mean ± Standard deviation (SD). severity scoring of atopic dermatitis (SCORD), thymus and activation-regulated chemokine (TARC).

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
