# Peer review of "Effect of the Lactococcus Lactis 11/19-B1 Strain on Atopic Dermatitis in a Clinical Test and Mouse Model"

_nutrients, 2020, doi:10.3390/nu12030763_

Round 1
Reviewer 1 Report
This is an interesting study by Suzuki et al. who demonstrate that the severity of atopic dermatitis (AD) in patients improved upon intake of yoghurt containing certain lactic acid bacteria (LAB). To investigate the species involved, the researches set up an AD mouse model by repeated application of 1-fluoro-2, 4-dinitrobenzene (DNFB) and feeding of different LAB strains. In addition to severity of AD-like skin lesions, also ear swelling, IgE levels, frequency of cytokine producing CD4+ T helper cell subsets, and eosinophil infiltration were scored. It was observed that L. lactis 11/19-B1 was the only strain that reduced parameters of AD in this mouse model. Reduced frequencies of all polarized CD4+ T cell subsets were observed but this could not be attributed to an increased frequency of regulatory T cells.
Some questions that remain are:
- Figure 1: Showing results of the SCORAD index in AD patients that ingested yoghurt without L. lactis 11/19-B1 would more convincingly demonstrate a beneficial effect this strain in improving AD pathology. Why were these subjects not included in the study ?
- Have the authors performed flow cytometric analysis of blood samples of patients before and after yoghurt intake and analyzed CD4+ T cells and cytokines production? If so, they should show the results of this analysis.
- Have tissue biopsies from patients been analyzed for infiltration of lymphocytes and eosinophils?
- In Figure 3 frequencies of cytokine producing CD4+ T cells are shown in PPs and CLNs. Are these the frequencies of the total CD4+ T cell pool ? Are the frequencies of total CD4+ T cell altered when related to total CD45+ cells ? What are the absolute numbers of cytokine producing CD4+ T cells in the different conditions ? Are these altered as well by L. lactis 11/19-B1 or control strain?
- The similar question should be addressed about the regulatory T cells. What are the absolute numbers of Tregs in PPs and CLNs ? Are these altered? And moreover, is the functionality (i.e. suppressive activity) of the Tregs altered when mice are fed L. lactis 11/19-B1?
- Innate lymphoid cells (ILC) have been implicated in AD pathology. Did the authors analyze ILC frequencies and/or absolute cell numbers in PPs and CLNs ?
Author Response
Thank you very much for your kind comments.
Comment #1
-Figure 1: Showing results of the SCORAD index in AD patients that ingested yoghurt without L. lactis 11/19-B1 would more convincingly demonstrate a beneficial effect this strain in improving AD pathology. Why were these subjects not included in the study ?
Response
We agree with the reviewer regarding this point. Initially, we also planned to investigate yogurt without Lactococcus lactis 11/19-B1 strain as a control. However, we gave up on this as we could not recruit a sufficient number of atopic dermatitis patients despite collecting volunteers at four clinics. Moreover, one pediatrician refused to assist with a poorly effective control study. Due to these limitations, we designed mouse experiments. We will try to carry out a double-blind controlled study to estimate the beneficial effects of the 11/19-B1 strain in future to answer this question.
Comment #2
- Have the authors performed flow cytometric analysis of blood samples of patients before and after yoghurt intake and analyzed CD4+ T cells and cytokines production? If so, they should show the results of this analysis.
- Have tissue biopsies from patients been analyzed for infiltration of lymphocytes and eosinophils?
Response
This study was carried out at clinics in a town offering general pediatric care. Therefore, we could not collect samples other than serum. Therefore, we carried out a mouse atopic model for detailed study instead of clinical tests.
Comment #3
- In Figure 3 frequencies of cytokine producing CD4+ T cells are shown in PPs and CLNs. Are these the frequencies of the total CD4+ T cell pool ? Are the frequencies of total CD4+ T cell altered when related to total CD45+ cells ? What are the absolute numbers of cytokine producing CD4+ T cells in the different conditions ? Are these altered as well by L. lactis 11/19-B1 or control strain?
Response
Thank you very much for your appropriate suggestion. The frequencies of cytokine-producing CD4+ were the frequencies of total CD4+. It was best that we showed the frequencies as the absolute number of total CD45+ cells. However, the recovery rate of lymphocytes from lymph nodes was not constant and the number of recovered lymphocytes from lymph nodes was insufficient for several analyses. So, we could measure only the frequencies of cytokine-producing CD4+ among total CD4+. To clarity, the vertical axis of Figure 3 was revised from XX cells (%) to XX cells in CD4+ (%). The title of Figure 3 was also revised.
Comment #4
- The similar question should be addressed about the regulatory T cells. What are the absolute numbers of Tregs in PPs and CLNs ? Are these altered? And moreover, is the functionality (i.e. suppressive activity) of the Tregs altered when mice are fed L. lactis11/19-B1?
Response
We could not count the absolute numbers of Tregs in PPs and CLNs for the same reason as mentioned above. We recovered lymphocytes from lymph nodes by rubbing the lymph nodes onto mesh. Through this procedure we could not estimate the recovery rate of lymphocytes from lymph nodes. Thus, we could not determine the absolute cell number.
In this study, we observed only the phenomenon of atopy remission. We could not observe Tregs functionality.
Comment #5
- Innate lymphoid cells (ILC) have been implicated in AD pathology. Did the authors analyze ILC frequencies and/or absolute cell numbers in PPs and CLNs ?
Response
We did not analyze ILC. The effect of 11/19-B1 strains on innate immunity is an interesting point, so we will study this as our next project. Thank you very much for your suggestion.
Reviewer 2 Report
In the paper entitled “Effect of the Lactococcus lactis 11/19-B1 strain on atopic dermatitis (AD) in a clinical test and mouse model”, the authors show that ingestion of yogurt fermented with various bacterial strains for 8 weeks significantly improved the severity scoring of atopic dermatitis in patients. They also show that the effect is likely mediated prevalently by L. lactis 11/19-B1, as determined in an AD model induced in mice by administration of 1-fluoro-2, 4-dinitrobenzene (DNFB).
Major points
-The authors declare that patients are under treatment. The authors must indicate the type of treatment for each patient and the statistical analysis should be corrected for this variable.
-The statistical analysis part should be improved. Have the authors check whether the data are normally distributed? This is particularly important for the histological part.
-How is the histological analysis performed? How many sections? What area has been analyzed? Mean or median value of the sections analyzed are considered for the statistical analysis?
-It would be interesting to evaluate whether a mix of all the lactic acid bacteria contained in the yogurt may exert additive/synergic effects in the AD model
Author Response
Thank you very much for your important comments.
Comment #1
-The authors declare that patients are under treatment. The authors must indicate the type of treatment for each patient and the statistical analysis should be corrected for this variable.
Response
Thank you very much for your suggestion. All subjects received topical steroids and one or more additional drugs. Almost all subjects continued these treatments during the study, but a few subjects reduced drug use late in the yogurt-intake period because of improvement in symptoms. It was difficult to correct the data statistically because the treatments were various. However, no subject began a new treatment. Therefore, we have concluded that the remission of symptoms was caused by the yogurt intake.
The above content was added in Materials and Methods (p. 2, l.76-78), Table 1 and Discussion (p.10, l.263-265).
Comment #2
-The statistical analysis part should be improved. Have the authors check whether the data are normally distributed? This is particularly important for the histological part.
Response
Thank you very much for your suggestion. We noticed some inappropriate analyses after reviewing the whole statistical analyses according your suggestion. All results were not normally distributed, so we should not apply a t-test. Therefore, all results were analyzed again by Mann-Whitney U test or Steel-Dwass test.
We revised Materials and Methods (p.4, l. 156-158), Fig. 1B, Fig. 2B, 2C, 2D, Fig. 3, Fig 4, Fig. 5 and their legends. Thank you very much for pointing out the mistake.
Comment #3
-How is the histological analysis performed? How many sections? What area has been analyzed? Mean or median value of the sections analyzed are considered for the statistical analysis?
Response
Our description was insufficient and, in accordance with the reviewer’s suggestion, we added some sentences in Materials and Methods (p.4, l.148-154) and legend for Fig. 5 (p.9, l.245-246)
Comment #4
-It would be interesting to evaluate whether a mix of all the lactic acid bacteria contained in the yogurt may exert additive/synergic effects in the AD model
Response
This is a very interesting and important suggestion. The test yogurt contained 3 species in addition to the 2 basic bacterial species for making yogurt, S. thermophiles and L. bulgaricus. Probably there are various synergistic effects between these species. It is a very complicated study, so we will try it after the development of simple evaluation method for in vitro use as our next project. Thank you very much for your interesting discussion.
Round 2
Reviewer 1 Report
While not all my questions were answered I understand these are difficult to perform in the setting of the researchers. I am satisfied with the overall results presented in the manuscript.
Reviewer 2 Report
The authors replied to my queries